# SDGs-Based River Health Assessment for Small- and Medium-Sized Watersheds

**Chenyang Xue, Chaofeng Shao ***  **and Sihan Chen**

College of Environmental Science and Engineering, Nankai University, Tianjin 300071, China; 2120170641@mail.nankai.edu.cn (C.X.); chensihan26@163.com (S.C.)

* Correspondence: shaocf@nankai.edu.cn

**Abstract:** A river health assessment index system was established, focusing on the realistic needs of county sustainable development and the refined management of small- and medium-sized watersheds. The index system takes into consideration the United Nations' Sustainable Development Goals (SDGs) and the vulnerability characteristics of small- and medium-sized watershed ecosystems and consists of 15 indicators in four areas: clean water, sanitation, the present status of biodiversity and threats to biodiversity. This paper uses the minimum discrimination information principle to construct a dynamic combination-weighting technology composed of a subjective weighting method (document frequency method) and an objective weighting method (entropy weight method). Using the fuzzy matter-element analysis theory, a comprehensive river health assessment technology system was constructed. Baoxing County was chosen as the research area and the results reveal that: (1) Key indicators are the biodiversity index of fish, water use intensity, endemic or indicative species retention, and chemical oxygen demand (COD) emissions. (2) The Euclid approach degree of Baoxing County indicates that the entire river is in a moderate state of health. In the future, towns must take targeted measures to coordinate the relationship between the ecological environment and socio-economic development, and enhancement and releasing must be prioritised.

**Keywords:** SDGs; river health assessment; small- and medium-sized watersheds; fuzzy matter-element model

## 1. Introduction

As the source of large rivers, small- and medium-sized watersheds are generally located in mountainous and hilly areas with high altitudes and complex terrain. They are characterised by poor basin regulation and storage capacity, short confluence time [1], high ecosystem vulnerability, susceptible to natural disasters and significantly affected by human activities. They are also counties with prominent contradictions between human populations and resources, and a vicious circle of economic poverty and ecological poverty. The counties located in small- and medium-sized watersheds urgently need high-level sustainable development pathways that integrate protection and development. According to Chinese statistics, counties located in ecologically vulnerable areas, such as water source areas, account for approximately 40% of all state-level poverty-stricken counties [2], and there are widespread problems with the over-exploitation of ecological resources. Hydropower development is an important means of water resource exploitation in small- and medium-sized watersheds, which poses a great threat to river ecosystems. Therefore, strengthening the management of water resources in small- and medium-sized watersheds is the key to the realisation of the sustainable utilisation of water resources and to ensuring the healthy and sustainable development of the national economy. River health assessment is a comprehensive assessment technology based on a quality judgement of natural

water ecosystems, which also considers the service function of water ecosystems and integrates social and economic factors. It is an important scientific basis for guiding a county's sustainable development.

At present, there are two main methods of river ecosystem health assessment in China and internationally: the indicative species method [3,4] and the index system method [5]. The indicative species method employs population diversity and richness indicators to reflect the health status of river ecosystems, using such indices as the Index of Biotic Integrity (IBI) and the Invertebrate Species Index (ISI) [6]. The index system method evaluates water health through influencing factors, such as physical, chemical, biological and socio-economic indicators at different scales. Representative methods include the score of Riparian, Channel and Environmental (RCE) [7] and the Index of Stream Condition (ISC) [8]. The index system method can make further evaluations using the comprehensive health index, fuzzy synthetic evaluation, grey clustering analysis and other models [9]. For example, Zhao et al. evaluated the health of an urban river ecosystem in Ningbo from five perspectives: water content, water quality, aquatic organisms, physical structure and riparian zone [10]. Based on natural and social attributes, Lin selected 20 indices to evaluate the river health of the Pearl River Basin [11].

In September 2015, 193 United Nations members formally adopted 17 sustainable development goals (SDGs). Among these, Target 6, 'Clean water and sanitation', and Target 14, 'Life below water', outline clear requirements for the sustainable development of water ecosystems [12]. In other words, the purpose of protection is for development. Sustainable and healthy water is dependent on coordination between human society and the natural environment, and the dialectical unity of protection and development. However, current river health assessment research only focuses on the natural health of water bodies and the human behaviour of pollution discharge, both of which are confined to the purpose of protecting the ecological environment [13,14]. The concern of such research for human activities is one-sided and does not meet the requirements of the SDGs that protection is as important as development. At the same time, such research exhibits strong subjectivity in the calculation of index weight. Therefore, based on the shortcomings of the river health assessment index system and evaluation model, and considering human behaviour in combination with the new requirements of sustainable development, this study constructs a river health assessment index system of small- and medium-sized watersheds based on the SDGs, using Baoxing County as an example, and establishes an evaluation model based on the Euclid approach degree according to fuzzy matter-element analysis theory. It uses a dynamic weighting method, based on the coupling of the entropy weight method and the Document Frequency (DF) method, to make a comprehensive and objective assessment of river health.

## 2. SDGs-Based Index System of River Health Assessment for Small- and Medium-Sized Watersheds

In accordance with the concept of human-oriented and harmonious symbiotic water management, sustainable development seeks to completely solve development problems in the three spheres of society, economy and environment between 2015 and 2030. Therefore, focusing on the ecological and social benefits of the river, river health assessment based on SDGs should be evaluated from the foothold of a dynamic equilibrium situation for natural ecology and human society. However, current research on the establishment of river health assessment index systems continues to contain a number of deficiencies: (1) In addition to providing habitat for aquatic organisms, rivers can provide drinking water resources, tourism landscapes, transportation, hydropower development, agricultural irrigation and other functions for human society. However, river health assessments internationally, and in some Chinese studies, reflect the advantages and disadvantages of the river's natural state, with little attention paid to the service function of rivers to humans [15]. Even when the socio-economic value of a river is taken into account, attention is often paid to human interference behaviours for the purposes of river protection, while the opportunities provided by the river for the sustainable development of human society are ignored [16]. (2) When evaluating the health status of underwater organisms, many index systems are too biased to distinguish the functions and biodiversity between

phytoplankton, zooplankton, benthos, fish, and indicative species [17]. (3) At present, river health assessment related to social attributes is biased towards human interference behaviours in the selection of indicators. For example, research only focuses on the impact of human pollution behaviour on the habitats of aquatic organisms (i.e., water quality) but fails to take into consideration the impact from other perspectives such as the deficiency of ecological flow.

　　Therefore, following the principles of integrity, hierarchy, rigor and parsimony, this study fully complies with the requirements of the 2030 sustainable development goal for river health assessment, focusing on both ecological protection and development behaviour. By considering the actual situation of the study area, and discussing the characteristics of small- and medium-sized river basins, an index system was established around sustainable development and management needs. The system is composed of four layers: the overall layer, the target layer, the status layer and the indicator layer (Figure 1). Overall layer: this is a high-level generalisation of the river health assessment index system of small- and medium-sized watersheds, indicating the overall comprehensive health status level. Target layer: Adopting as a goal the requirements for river health proposed by the SDGs, the evaluation of river health has two comprehensive goals: clean water and sanitation, and life below the water. Status layer: this sets state indicators under each goal. SDGs Target 6 can be divided into clean water and sanitation, while Target 14 can be evaluated from the current situation of underwater organisms and the potential threats they face. Indicator layer: This describes the different elements of each classification indicator, and directly reflects river health status using quantitative or qualitative indicators. The selection of specific indicators should be representative, independent, scientific, comprehensive, easy to obtain and practical. Therefore, it is proposed that the river health assessment system for small- and medium-sized watersheds, based on SDGs, should be composed of 15 indicators. On the basis of existing research, indicators related to development, such as centralised water supply ratio, and characteristic indicators of small- and medium-sized watersheds, such as the proportion of water reduced river reach, should be added. Of these, nine monitoring factors, including pH, Dissolved Oxygen (DO), Suspended Solids (SS), Chemical Oxygen Demand (COD), Biochemical Oxygen Demand ($BOD_5$), potassium permanganate index, ammonia nitrogen, Total Phosphorus (TP) and Total Nitrogen (TN) are used for the calculation of the Nemerow comprehensive pollution index, a multi-factor environmental quality index considering extremum [18]. The specific meaning of each indicator is shown in Table 1. Indicator direction indicates the correlation between the indicator and the SDGs. The indicator direction is positive when there is a positive correlation, and the indicator direction is negative when there is a negative correlation.

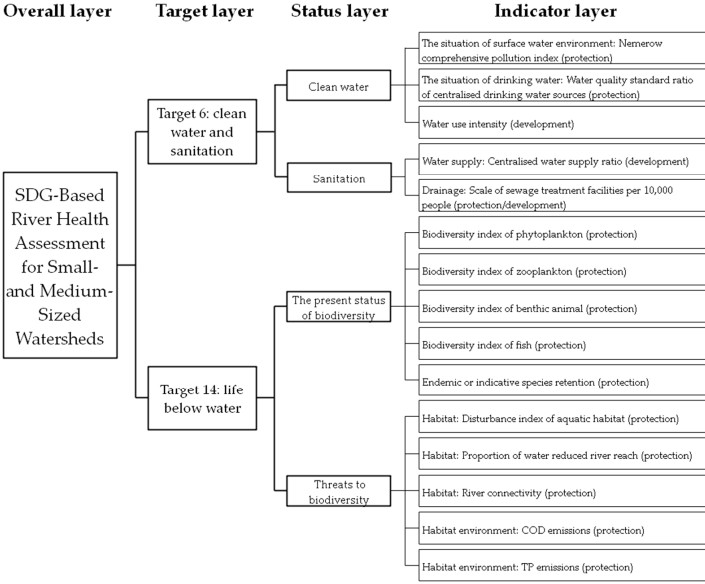

**Figure 1.** The sustainable development goals (SDGs)-based index system structure of river health assessment for small- and medium-sized watersheds.

**Table 1.** The meaning of SDGs-based index system of river health assessment for small- and medium-sized watersheds.

| Target | Status | Indicator | The Specific Meaning of Each Indicator | The Correlation with SDGs (Protection/Development) | Indicator Direction |
|---|---|---|---|---|---|
| Target 6: clean water and sanitation | Clean water | Nemerow comprehensive pollution index | The water environmental quality of the monitoring section is evaluated according to various chemical monitoring indicators, and the calculation formula is: $I_p = \sqrt{\frac{(I^i_{max})^2+(\bar{I})^2}{2}}$, where: $\bar{I} = \frac{1}{n}\sum_{i=1}^{n} I_i$, $I_i = \frac{C_i}{C_{oi}}$. In the formula, $I_p$ is the Nemerow index; $I^i_{max}$ is the maximum value of the pollution index in all evaluation factors; $\bar{I}$ is the average value of the pollution index in all evaluation factors; $I_i$ is the pollution index of factor i; $C_i$ is the monitoring value of factor i; $C_{oi}$ is the water quality standard value of factor i. The target water quality of Baoxing River is class II standard. | The larger the value, the lower the degree of protection, which is not conducive to sustainable development. | − |
| | | Water quality standard ratio of centralised drinking water sources | The proportion of the water intake of the centralised drinking water source in the assessment area that conforms to the drinking water quality (class I standard) to the total water intake. | The larger the value, the higher the degree of protection, which is conducive to sustainable development. | + |
| | | Water use intensity | The ratio is obtained by dividing the annual water withdrawal by the total annual renewable water resources. According to international practice, it is generally considered that the water use intensity cannot exceed 40%. | Under the premise of not exceeding the utilisation limit, the larger the value, the higher the degree of development, which is conducive to sustainable development. | + |
| | Sanitation | Centralised water supply ratio | The proportion of the population supplied by the centralised water supply project to the regional population. An indicator of water quality from the perspective of water supply sanitation facilities. | The larger the value, the higher the degree of development, which is conducive to sustainable development. | + |
| | | Scale of sewage treatment facilities per 10,000 people | The ratio of the total scale of sewage treatment facilities to the total population in the region, which is used to evaluate the regional sewage treatment capacity. An indicator of water quality from the perspective of drainage sanitation facilities. | The larger the value, the higher the degree of protection and development, which is conducive to sustainable development. | + |

**Table 1.** *Cont.*

| Target | Status | Indicator | The Specific Meaning of Each Indicator | The Correlation with SDGs (Protection/Development) | Indicator Direction |
|---|---|---|---|---|---|
| Target 14: life below water | The present status of biodiversity | Biodiversity index of phytoplankton | Shannon-Wiener index is used to characterise the biodiversity of phytoplankton in the river. The larger the index, the higher the complexity of the community. $H = -\sum_{i=1}^{r}(\frac{n_i}{N}ln\frac{n_i}{N})$, where $n_i$ represents the number of individuals of the species i, $N$ represents the total number of individuals of all species in the community. | The larger the value, the higher the degree of protection, which is conducive to sustainable development. | + |
| | | Biodiversity index of zooplankton | Shannon-Wiener index is used to characterise the biodiversity of zooplankton in the river. The larger the index, the higher the complexity of the community. | The larger the value, the higher the degree of protection, which is conducive to sustainable development. | + |
| | | Biodiversity index of benthic animal | Shannon-Wiener index is used to characterise the biodiversity of benthic animals in the river. The larger the index, the higher the complexity of the community. | The larger the value, the higher the degree of protection, which is conducive to sustainable development. | + |
| | | Biodiversity index of fish | Shannon-Wiener index is used to characterise the biodiversity of fish in the river. The larger the index, the higher the complexity of the community. | The larger the value, the higher the degree of protection, which is conducive to sustainable development. | + |
| | | Endemic or indicative species retention | This indicator reflects the protection extent of river endemic, indicative species and endangered species. According to the data obtained from the water biology survey or questionnaire statistics, and compared with the historical information, the value is calculated by the grading assignment method (the full score of 100). | The larger the value, the higher the degree of protection, which is conducive to sustainable development. | + |
| | Threats to biodiversity | Disturbance index of aquatic habitat | This indicator reflects the impact of human activities such as sand digging and shipping on water habitat. When there is an invasive phenomenon of alien species in the water ecosystem, it is necessary to add the assessment of the invasive ratio of alien species. Based on the data obtained from the field survey, the value is calculated by the grading assignment method (the full score of 100). | The larger the value, the higher the degree of protection, which is conducive to sustainable development. | + |
| | | Proportion of water reduced river reach | This indicator reflects the proportion of the length of the water reducing reach to the total length of the river in the assessment area, the higher the value, the greater the damage to the habitat of aquatic organisms. | The larger the value, the lower the degree of protection, which is not conducive to sustainable development. | − |
| | | River connectivity | Due to the interference of human activities, especially the construction of hydropower stations, dams and other water conservancy projects, the continuity of the river between upstream and downstream is interrupted, which has a negative impact on its self-purification capacity and biological migration channel. This indicator uses the number of dams per 100 km of river to evaluate the connectivity of the river. | The larger the value, the lower the degree of protection, which is not conducive to sustainable development. | − |
| | | COD emissions | COD is generally used to indicate the content of organic matter in wastewater, which reflects the pollution degree of organic matter in water. The higher the COD value, the heavier the organic pollutant pollution in the water, and the greater the threat to underwater life. | The larger the value, the lower the degree of protection, which is not conducive to sustainable development. | − |
| | | TP emissions | This indicator characterises the degree of eutrophication of water bodies. The higher the value, the more serious the impact on underwater organisms. | The larger the value, the lower the degree of protection, which is not conducive to sustainable development. | − |

## 3. River Health Assessment Model

### *3.1. Determination of Combination Weight*

In the process of river health assessment, reasonably determining the weight for scientific assessment is essential. At present, there are three main methods to determine the weight of evaluation factors: subjective weights, objective weights and combined weights. Typical subjective weighting methods include the Delphi method and the Analytic Hierarchy Process (AHP), which comprehensively consider the subjective experience and knowledge levels of decision-makers, and can reduce or eliminate the impact of data errors on evaluation results. Typical objective weighting methods include Principal Component Analysis, the entropy weight method, and the coefficient of variation method [19]. These are based on mathematical theory and are more scientific, but deviation from the actual situation is easy due to the lack of subjective control. In the application process of the traditional fuzzy matter-element analysis model, most scholars adopt single weighting methods, such as AHP [20,21], the entropy weight method [22,23] and the variation coefficient method [24]. In recent years, many scholars have coupled the characteristics of the two methods and adopted a combined weighting method of subjective and objective weights in order to solve the bias of the single weighting method [25].

Recent years have also seen the emergence of web text mining technology, which is the research and practice of using computer linguistics, statistical analysis and other principles to extract user demand information from web text. This can be applied to web browsing, text retrieval, classification, clustering, association analysis, document summaries, trend prediction and so on. In automatic text categorisation, the Term Frequency-Inverse Document Frequency (TF-IDF) algorithm is the most commonly used weighting method in evaluating the importance of words in a document set or corpus [26]. The importance of a word increases in proportion to the number of times it appears in a document, but decreases in inverse proportion to the number of times it appears in a corpus. In this paper, the algorithm is introduced into the index weighting model of the assessment system in order to undertake the subjective weighting method. The weight is set according to the occurrence frequency of terms or keywords. In other words, in this subjective weighting method, the importance degree is reflected in the subjective research willingness of researchers, their research interest and social attention to relevant content. However, with the help of internet tools and the full use of data mining technology, the scope of information acquisition is extensive and comprehensive, and information can be analysed faster. With the continuous scientific update of data, the determination of weights has become more dynamic, scientific, authoritative, reliable and convenient than traditional subjective weighting methods.

Therefore, on the basis of the above review of methods, this paper combines the advantages of subjective and objective weighting methods to present an improved weighting method, which uses the minimum discrimination information principle to calculate the combined weights of DF weights and entropy weights.

### 3.1.1. Subjective Index Weighting Method Using Document Frequency Weights

The traditional TF-IDF algorithm is used to calculate the weight of different keywords in a document in order to measure and grade the association degree of the document. If this algorithm is introduced into the index weighting model of the assessment system, only the frequency of documents containing keywords needs to be considered, as follows:

$$DF_j = \frac{N(j)}{N} \tag{1}$$

Where, $N$ represents the total number of documents in the database, and $N(j)$ represents the number of documents in the document set containing entry $j$.

The weight of indicator $j$ is:

$$w_{aj} = \frac{DF_j}{\sum_{j=1}^{n} DF_j} = \frac{N(j)}{\sum_{j=1}^{n} N(j)} \tag{2}$$

### 3.1.2. Objective Index Weighting Method Using Entropy Weights

In information systems, the uncertainty of signals in communication processes is known as 'information entropy' [27]. The entropy weight method is an objective weighting method, which reflects the variation degree of each index. If the entropy value of an indicator is lower, the degree of variation of the indicator is larger, the more information is provided, the greater the role it plays in the comprehensive evaluation, and the higher its weight should be. The steps taken in the entropy weight method to determine the weight of indicators are as follows [28]:

(1) Construction of initial decision matrix

Supposing there are $m$ objects to evaluate $A = \{A_1, A_2, \ldots, A_m\}$, and $n$ evaluation indices $C = \{C_1, C_2, \ldots, C_n\}$. The value of the index $C_j$ of object $A_i$ is recorded as $x_{ij}$. $X = (x_{ij})_{n \times m}$ is the initial decision matrix of the evaluation issue.

(2) Non-dimensional processing of data

$$y_{ij} = \frac{x_{ij} - min(x_j)}{max(x_j) - min(x_j)} \tag{3}$$

$$y_{ij} = \frac{max(x_j) - x_{ij}}{max(x_j) - min(x_j)} \tag{4}$$

where, $y_{ij}$ is the normalised value (non-dimensional value) of the index $C_j$ of object $A_i$. $max(x_j)$ and $min(x_j)$ represents the maximum and minimum value of the eigenvalues of each evaluation index. When the indices are 'forward-type', the calculation for normalisation uses Equation (3). When the indices are 'reverse-type', the calculation for normalisation uses Equation (4).

(3) Calculating entropy value

$$E_j = -\frac{1}{\ln(m)} \left( \sum_{i=1}^{m} p_{ij} \ln(p_{ij}) \right) \tag{5}$$

In Equation (5), $p_{ij} = \frac{1 + y_{ij}}{\sum_{i=1}^{m}(1 + y_{ij})}$.

(4) Calculation of index weights

$$w_{bj} = \frac{1 - E_j}{\sum_{j=1}^{n}(1 - E_j)} \tag{6}$$

### 3.1.3. Combination Weighting Method Based on the Minimum Discrimination Information Principle

In order that the index weight not only includes the subjective judgement of the decision-maker, but also is constrained by objective conditions, the distance between the combined weight $w_j$, the subjective weight $w_{aj}$ and the objective weight $w_{bj}$ should be as close as possible. In order to achieve this goal, using the minimum discrimination information principle, the following objective functions are established [19]:

$$minF = \sum_{j=1}^{n} w_j ln\left(\frac{w_j}{w_{aj}}\right) + \sum_{j=1}^{n} w_j ln\left(\frac{w_j}{w_{bj}}\right) \tag{7}$$

$$s.t. \sum_{j=1}^{n} w_j = 1; w_j > 0 \tag{8}$$

Lagrange multiplier is then introduced to solve the above problems:

$$w_j = \frac{\sqrt{w_{aj}w_{bj}}}{\sum_{j=1}^{n}\sqrt{w_{aj}w_{bj}}} \tag{9}$$

### 3.2. Fuzzy Matter-Element Model

As shown in Figure 2, obtaining the Euclid approach degree through a fuzzy matter-element model is as follows:

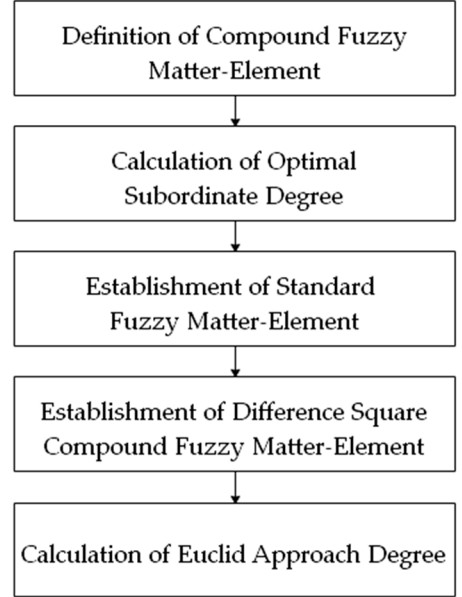

**Figure 2.** The assessment process based on the fuzzy matter-element model.

### 3.2.1. The Concept Of Fuzzy Matter-Element

Given the name of matter $M$, its value of feature $C$ is $X$. The basic elements of things, or matter-element for short, are described according to the tripled order of 'matter, feature, value'. In other words, $R = (M, C, X)$. If $X$ is fuzzy, it is called fuzzy matter-element. If a matter has n features $(C_1, C_2, \cdots, C_n)$ and corresponding fuzzy values $(X_1, X_2, \cdots, X_n)$, $R$ is $n$ dimensions fuzzy matter-element, also denoted by $R = (M, C, X)$. $m$ matters with $n$ dimensions matter-element form composite matter-element $R_{mn}$. If values of $R_{mn}$ are fuzzy, it is known as $n$ dimensions composite fuzzy matter-element, denoted by [24,29]:

$$R_{mn} = \begin{bmatrix} & M_1 & M_2 & \cdots & M_m \\ C_1 & X_{11} & X_{21} & \cdots & X_{m1} \\ C_2 & X_{12} & X_{22} & \cdots & X_{m2} \\ \vdots & \vdots & \vdots & & \vdots \\ C_n & X_{1n} & X_{2n} & \cdots & X_{mn} \end{bmatrix} \tag{10}$$

where, $R_{mn}$ denotes the $n$ dimensions composite fuzzy matter-element of the m-matching schemes; $M_i$ is matter $i$, and $i = 1, 2, \cdots, m$; $C_j$ is feature $j$, and $j = 1, 2, \cdots, n$; $X_{ij}$ is the fuzzy value of matter $i$ on feature $j$.

### 3.2.2. Calculation of Optimal Subordinate Degree

$$
R_{mn} = \begin{bmatrix}
 & M_1 & M_2 & \cdots & M_m \\
C_1 & \mu_{11} & \mu_{21} & \cdots & \mu_{m1} \\
C_2 & \mu_{12} & \mu_{22} & \cdots & \mu_{m2} \\
\vdots & \vdots & \vdots & & \vdots \\
C_n & \mu_{1n} & \mu_{2n} & \cdots & \mu_{mn}
\end{bmatrix}
\tag{11}
$$

Subordinate degree refers to the degree of membership corresponding to each evaluation index from the corresponding index value of the standard sample. $\mu_{ij}$ can be defined by the principle of optimal subordinate degree [22].

Forward index subordinate degree calculation:

$$
\mu_{ij} = \frac{X_{ij} - minX_{ij}}{maxX_{ij} - minX_{ij}}
\tag{12}
$$

Reverse index subordinate degree calculation:

$$
\mu_{ij} = \frac{maxX_{ij} - X_{ij}}{maxX_{ij} - minX_{ij}}
\tag{13}
$$

### 3.2.3. Construction of the Standard Fuzzy Matter-Element Matrix And the Difference Square Compound Fuzzy Matter-Element Matrix

$$
R_{0n} = \begin{bmatrix}
 & M_0 \\
C_1 & \mu_{01} \\
C_2 & \mu_{02} \\
\vdots & \vdots \\
C_n & \mu_{0n}
\end{bmatrix}
\tag{14}
$$

where $R_{0n}$ represents the standard fuzzy matter-element; and $\mu_{0n}$ represents the maximum or minimum value of the subordinate degree.

$$
R_\Delta = \begin{bmatrix}
 & M_1 & M_2 & \cdots & M_m \\
C_1 & \Delta_{11} & \Delta_{21} & \cdots & \Delta_{m1} \\
C_2 & \Delta_{12} & \Delta_{22} & \cdots & \Delta_{m2} \\
\vdots & \vdots & \vdots & & \vdots \\
C_n & \Delta_{1n} & \Delta_{2n} & \cdots & \Delta_{mn}
\end{bmatrix}
\tag{15}
$$

In the formula, $\Delta_{ij} = (\mu_{0j} - \mu_{ij})^2$; $R_\Delta$ represents the difference square compound fuzzy matter-element [30].

### 3.2.4. Calculation of the Euclid Approach Degree and Determination of River Health Status

(1) Quantitative evaluation

At last, the Euclid approach degree, which is used to describe the similarity between the assessed product and the standard product, can be evaluated. A large value implies a high correlation degree

between the two samples, whereas a small value implies a low correlation degree between the two samples. It is expressed as [20]:

$$\rho H_i = 1 - \sqrt{\sum_{j=1}^{n} w_j \Delta_{ij}} \, (i = 1, 2, \cdots, m, j = 1, 2, \cdots, n) \tag{16}$$

The Euclid approach degree of compound fuzzy matter-element is:

$$R_{\rho H} = \begin{bmatrix} & M_1 & M_2 & \cdots & M_m \\ \rho H_i & \rho H_1 & \rho H_2 & \cdots & \rho H_m \end{bmatrix} \tag{17}$$

(2) Qualitative evaluation

According to the quantitative evaluation results, the evaluation grade is divided to qualitatively describe the health of the river. However, this study does not use the standard gradational method [31], but rather the Jenks Natural Breaks for statistics. Using the data itself as a standard for graded evaluation, the relative health of county rivers can be derived, avoiding all 'excellent' or all 'extremely poor' results. The Jenks Natural Breaks algorithm was introduced in 1977 as a method to achieve optimal data classification. It uses algorithms such as 'sum of squared deviations' to divide a dataset into a certain number of homogenous classes. This method is commonly used in geographic information systems applications [32].

## 4. Research area and data sources

Baoxing River (Figure 3) is the primary source of the Qingyi River, a tributary of the Yangtze River. The total area of the basin is 3010 km$^2$ and the length of the water system is 104.8 km. Situated in the Hengduan Mountains of China, it is an area of extremely rich biological diversity. As the place where pandas were first discovered, Baoxing County not only belongs to the Sichuan-Yunnan forest and a biodiverse ecological functional area, it is also of great significance for the protection of the fauna and flora gene pool. As such, more than 90% of its territory has been incorporated into Giant Panda National Park. However, mining and hydropower development are the two major pillar industries in the area, contributing more than 55% to the GDP of the region. Baoxing County faces sharp contradictions and conflicts with regard to protection and development. Coordinating the relationship between ecological environmental protection and economic development is of great significance to the region's sustainable development.

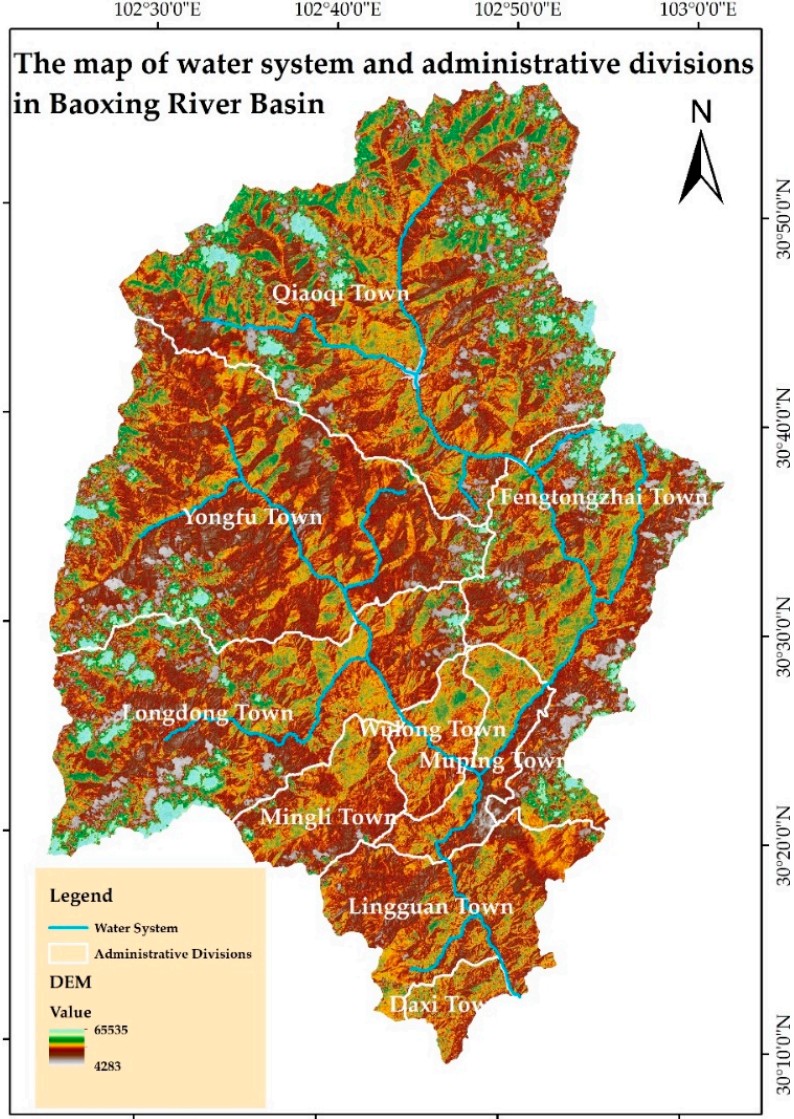

**Figure 3.** The map of water system and administrative divisions in Baoxing River Basin.

The plan to build the Giant Panda National Park was approved by the Chinese government in 2017. The year 2017 is the boundary between full protection and partial protection. Therefore, Baoxing County was selected as the research area and 2017 was selected as the reference year to evaluate. The required data comes from a field sampling survey, data sorting and statistical data analysis. Nine monitoring sections were set in nine towns to investigate the water quality and ecology quality. The local statistical yearbook was the major source of statistical data. The original data for each evaluation indicator are shown in Table 2.

**Table 2.** Original river health assessment data in the towns of Baoxing County.

| Town / Indicator | Muping Town | Lingguan Town | Longdong Town | Fengtongzhai Town | Qiaoqi Town | Yongfu Town | Mingli Town | Wulong Town | Daxi Town |
|---|---|---|---|---|---|---|---|---|---|
| Nemerow comprehensive pollution index $I_p$ | 0.61568 | 1.03069 | 0.57797 | 0.51569 | 0.56892 | 0.95416 | 0.34177 | 0.60711 | 0.61513 |
| Water quality standard ratio of Centralised drinking water sources (%) | 100 | 100 | 100 | 100 | 100 | 100 | 100 | 100 | 100 |
| Water use intensity (%) | 0.02000 | 0.27690 | 0.63310 | 0.05620 | 0.07040 | 0.03260 | 0.08310 | 0.19470 | 0.00070 |
| Centralised water supply ratio (%) | 1.00000 | 0.94476 | 0.40341 | 0.66143 | 0.22055 | 0.39707 | 0.13864 | 0.62914 | 0.31830 |
| Scale of sewage treatment facilities per 10,000 people (m³/d) | 2641.73096 | 2217.40580 | 189.57346 | 516.74246 | 845.43282 | 1306.16510 | 962.77279 | 496.68874 | 530.50398 |
| Biodiversity index of phytoplankton | 0.01100 | 0.01700 | 0.03000 | 0.02000 | 0.05800 | 0.02200 | 0.06000 | 0.02800 | 0.01700 |
| Biodiversity index of zooplankton | 0.00000 | 0.00000 | 0.32500 | 0.69300 | 0.95200 | 0.63700 | 0.00000 | 0.00000 | 0.00000 |
| Biodiversity index of benthic animal | 0.67300 | 0.45800 | 0.00000 | 1.63700 | 0.38500 | 0.95600 | 0.56200 | 0.45100 | 0.45800 |
| Biodiversity index of fish | 2.48491 | 1.38629 | 1.79176 | 2.48491 | 1.79176 | 1.38629 | 2.19722 | 2.19722 | 0.69315 |
| Endemic or indicative species retention | 62.00000 | 52.00000 | 38.00000 | 58.00000 | 68.00000 | 75.00000 | 55.00000 | 49.00000 | 85.00000 |
| Disturbance index of aquatic habitat | 100.00000 | 86.00000 | 88.00000 | 91.20000 | 100.00000 | 100.00000 | 100.00000 | 90.00000 | 100.00000 |
| Proportion of water reduced river reach (%) | 0.87964 | 0.14115 | 0.13978 | 0.33442 | 0.03380 | 0.05062 | 0.09849 | 0.54810 | 0.41629 |
| River connectivity (pcs/100km) | 15.60000 | 30.20000 | 52.60000 | 20.50000 | 3.30000 | 4.60000 | 15.80000 | 18.20000 | 0.00000 |
| COD emissions (t/a) | 84.55194 | 243.68873 | 25.16305 | 13.68964 | 43.84733 | 13.40036 | 16.77631 | 24.78215 | 17.97967 |
| TP emissions (t/a) | 0.79146 | 2.32724 | 0.18981 | 0.06512 | 0.38515 | 0.10964 | 0.13594 | 0.19378 | 0.14918 |

## 5. Result and Analysis

*5.1. Results*

(1) Determination of difference square compound fuzzy matter-element

A compound fuzzy matter-element $R_{mn}$ was constructed to assess the health of the Baoxing River according to each town's evaluation index data. The water quality standard ratio for centralised drinking water sources for every town was 100%, which does not constitute a fuzzy condition and has no impact on the assessment results. This factor was, therefore, eliminated from the evaluation. A fuzzy matter-element of the optimal subordinate degree was established from Equations (10)–(14). Based on relevant literature, this paper uses the maximum value to form the standard fuzzy matter-element, with the optimal subordinate degrees of all factors being equal to 1. According to Equation (15), the difference square compound fuzzy matter-element $R_\Delta$ can be obtained.

$$
\text{fuzzy matter} - \text{element of optimal subordinate degree } R_{mn} =
\begin{array}{c|ccccccccc}
 & \begin{array}{c}\text{Muping}\\\text{Town}\end{array} & \begin{array}{c}\text{Lingguan}\\\text{Town}\end{array} & \begin{array}{c}\text{Longdong}\\\text{Town}\end{array} & \begin{array}{c}\text{Fengtongzhai}\\\text{Town}\end{array} & \begin{array}{c}\text{Qiaoqi}\\\text{Town}\end{array} & \begin{array}{c}\text{Yongfu}\\\text{Town}\end{array} & \begin{array}{c}\text{Mingli}\\\text{Town}\end{array} & \begin{array}{c}\text{Wulong}\\\text{Town}\end{array} & \begin{array}{c}\text{Daxi}\\\text{Town}\end{array}\\
C_1 & 0.60240 & 0.00000 & 0.65715 & 0.74755 & 0.67027 & 0.11109 & 1.00000 & 0.61484 & 0.60320\\
C_2 & 0.03052 & 0.43675 & 1.00000 & 0.08776 & 0.11022 & 0.05044 & 0.13030 & 0.30677 & 0.00000\\
C_3 & 1.00000 & 0.93587 & 0.30739 & 0.60694 & 0.09509 & 0.30003 & 0.00000 & 0.56945 & 0.20858\\
C_4 & 1.00000 & 0.82696 & 0.00000 & 0.13342 & 0.26746 & 0.45535 & 0.31531 & 0.12524 & 0.13903\\
C_5 & 0.00000 & 0.12245 & 0.38776 & 0.18367 & 0.95918 & 0.22449 & 1.00000 & 0.34694 & 0.12245\\
C_6 & 0.00000 & 0.00000 & 0.34139 & 0.72794 & 1.00000 & 0.66912 & 0.00000 & 0.00000 & 0.00000\\
C_7 & 0.41112 & 0.27978 & 0.00000 & 1.00000 & 0.23519 & 0.58400 & 0.34331 & 0.27550 & 0.27978\\
C_8 & 1.00000 & 0.38685 & 0.61315 & 1.00000 & 0.61315 & 0.38685 & 0.83944 & 0.83944 & 0.00000\\
C_9 & 0.51064 & 0.29787 & 0.00000 & 0.42553 & 0.63830 & 0.78723 & 0.36170 & 0.23404 & 1.00000\\
C_{10} & 1.00000 & 0.00000 & 0.14286 & 0.37143 & 1.00000 & 1.00000 & 1.00000 & 0.28571 & 1.00000\\
C_{11} & 0.00000 & 0.87309 & 0.87470 & 0.64459 & 1.00000 & 0.98011 & 0.92352 & 0.39196 & 0.54780\\
C_{12} & 0.70342 & 0.42586 & 0.00000 & 0.61027 & 0.93726 & 0.91255 & 0.69962 & 0.65399 & 1.00000\\
C_{13} & 0.69103 & 0.00000 & 0.94892 & 0.99874 & 0.86779 & 1.00000 & 0.98534 & 0.95058 & 0.98011\\
C_{14} & 0.67891 & 0.00000 & 0.94488 & 1.00000 & 0.85853 & 0.98032 & 0.96870 & 0.94313 & 0.96284
\end{array}
\tag{18}
$$

$$
R_\Delta =
\begin{array}{c|ccccccccc}
 & \begin{array}{c}\text{Muping}\\\text{Town}\end{array} & \begin{array}{c}\text{Lingguan}\\\text{Town}\end{array} & \begin{array}{c}\text{Longdong}\\\text{Town}\end{array} & \begin{array}{c}\text{Fengtongzhai}\\\text{Town}\end{array} & \begin{array}{c}\text{Qiaoqi}\\\text{Town}\end{array} & \begin{array}{c}\text{Yongfu}\\\text{Town}\end{array} & \begin{array}{c}\text{Mingli}\\\text{Town}\end{array} & \begin{array}{c}\text{Wulong}\\\text{Town}\end{array} & \begin{array}{c}\text{Daxi}\\\text{Town}\end{array}\\
C_1 & 0.15809 & 1.00000 & 0.11755 & 0.06373 & 0.10872 & 0.79016 & 0.00000 & 0.14835 & 0.15745\\
C_2 & 0.93989 & 0.31725 & 0.00000 & 0.83218 & 0.79172 & 0.90166 & 0.75638 & 0.48057 & 1.00000\\
C_3 & 0.00000 & 0.00411 & 0.47971 & 0.15450 & 0.81886 & 0.48996 & 1.00000 & 0.18538 & 0.62634\\
C_4 & 0.00000 & 0.02994 & 1.00000 & 0.75096 & 0.53661 & 0.29664 & 0.46880 & 0.76520 & 0.74126\\
C_5 & 1.00000 & 0.77010 & 0.37484 & 0.66639 & 0.00167 & 0.60142 & 0.00000 & 0.42649 & 0.77010\\
C_6 & 1.00000 & 1.00000 & 0.43377 & 0.07402 & 0.00000 & 0.10948 & 1.00000 & 1.00000 & 1.00000\\
C_7 & 0.34678 & 0.51872 & 1.00000 & 0.00000 & 0.58494 & 0.17306 & 0.43124 & 0.52489 & 0.51872\\
C_8 & 0.00000 & 0.37595 & 0.14966 & 0.00000 & 0.14966 & 0.37595 & 0.02578 & 0.02578 & 1.00000\\
C_9 & 0.23947 & 0.49298 & 1.00000 & 0.33001 & 0.13083 & 0.04527 & 0.40742 & 0.58669 & 0.00000\\
C_{10} & 0.00000 & 1.00000 & 0.73469 & 0.39510 & 0.00000 & 0.00000 & 0.00000 & 0.51020 & 0.00000\\
C_{11} & 1.00000 & 0.01611 & 0.01570 & 0.12632 & 0.00000 & 0.00040 & 0.00585 & 0.36971 & 0.20448\\
C_{12} & 0.08796 & 0.32964 & 1.00000 & 0.15189 & 0.00394 & 0.00765 & 0.09023 & 0.11972 & 0.00000\\
C_{13} & 0.09546 & 1.00000 & 0.00261 & 0.00000 & 0.01748 & 0.00000 & 0.00021 & 0.00244 & 0.00040\\
C_{14} & 0.10310 & 1.00000 & 0.00304 & 0.00000 & 0.02001 & 0.00039 & 0.00098 & 0.00323 & 0.00138
\end{array}
\tag{19}
$$

(2) Determination of weights

The China National Knowledge Infrastructure was selected as the database, using full text as the search term. The following searches were entered: 'Nemerow Index || Water Pollution Index', 'Water Resources Utilisation || Water Resources Utilisation Intensity', 'Central Water Supply Rate || Water Supply Rate', 'Sewage Treatment Facility Scale || Sewage Treatment Plant Scale', 'Phytoplankton && Biodiversity', 'Zooplankton && Biodiversity', 'Benthos && Biodiversity', 'Fish && Biodiversity', 'Endemic species || Endemic organisms || Indicating organisms || Indicating species', '(Aquatic environment && Disturbance) || (River ecology && Disturbance) || (Aquatic environment && Disturbance)', 'Water reduced river reach || Water dehydrated river reach', 'Channel connectivity || Channel continuity || River connectivity || River continuity || Habitat integrity', 'COD emissions || Chemical oxygen demand Emissions' and 'TP Emissions || Total Phosphorus Emissions'. The publication date covered the twenty years from December 1, 1999 to December 1, 2019, and the literature classification choice was 'Mathematics/Physics/Mechanics/Astronomy', 'Chemistry/Metallurgy/Environment/Mine Industry', 'Agriculture' and 'Economics & Management'. Based on retrieval results, weights were dynamically determined using Equations (1)–(9) in combination with the entropy weight method (Table 3).

**Table 3.** The weight of each indicator.

| Indicator | Number of Articles Retrieved | DF Weights $w_{aj}$ | Entropy Weights $w_{bj}$ | Combined Weights $w_j$ |
|---|---|---|---|---|
| Nemerow comprehensive pollution index ($C_1$) | 4797 | 0.02547 | 0.05855 | 0.02547 |
| Water use intensity ($C_2$) | 31130 | 0.16529 | 0.08179 | 0.16529 |
| Centralised water supply ratio ($C_3$) | 1295 | 0.00688 | 0.08169 | 0.00688 |
| Scale of sewage treatment facilities per 10,000 people ($C_4$) | 2810 | 0.01492 | 0.08211 | 0.01492 |
| Biodiversity index of phytoplankton ($C_5$) | 15619 | 0.08293 | 0.09134 | 0.08293 |
| Biodiversity index of zooplankton ($C_6$) | 11836 | 0.06285 | 0.12239 | 0.06285 |
| Biodiversity index of benthic animal ($C_7$) | 10671 | 0.05666 | 0.05378 | 0.05666 |
| Biodiversity index of fish ($C_8$) | 51616 | 0.27406 | 0.06012 | 0.27406 |
| Endemic or indicative species retention ($C_9$) | 24388 | 0.12949 | 0.05807 | 0.12949 |
| Disturbance index of aquatic habitat ($C_{10}$) | 8569 | 0.04550 | 0.10057 | 0.04550 |
| Proportion of water reduced river reach ($C_{11}$) | 743 | 0.00395 | 0.05819 | 0.00395 |
| River connectivity ($C_{12}$) | 1920 | 0.01019 | 0.05102 | 0.01019 |
| COD emissions ($C_{13}$) | 20753 | 0.11019 | 0.05048 | 0.11019 |
| TP emissions ($C_{14}$) | 2188 | 0.01162 | 0.04990 | 0.01162 |

The values of $w_j$ and $R_\Delta$ are substituted into Equation (16) to calculate the Euclid approach degree of each town.

$$R_{\rho H} = \begin{bmatrix} & \text{Muping} & \text{Lingguan} & \text{Longdong} & \text{Fengtongzhai} & \text{Qiaoqi} & \text{Yongfu} & \text{Mingli} & \text{Wulong} & \text{Daxi} \\ & \text{Town} & \text{Town} & \text{Town} & \text{Town} & \text{Town} & \text{Town} & \text{Town} & \text{Town} & \text{Town} \\ \rho H_i & 0.38977 & 0.24363 & 0.40777 & 0.47644 & 0.50938 & 0.40628 & 0.46430 & 0.42286 & 0.21533 \end{bmatrix} \quad (20)$$

According to the calculated Euclid approach degree, the river health in the towns of Baoxing County can be listed from high to poor level: Qiaoqi Town, Fengtongzhai Town, Mingli Town, Wulong Town, Longdong Town, Yongfu Town, Muping Town, Lingguan Town and Daxi Town. The Jenks Natural Breaks method was used to classify the river health of these towns, and the results show that the water bodies of Qiaoqi Town, Fengtongzhai Town and Mingli Town are in a high state of health,

the water bodies of Wulong Town are in a good state of health, the water bodies of Longdong Town, Yongfu Town, and Muping Town are in a moderate state of health, and the water bodies of Lingguan Town and Daxi Town are in a poor state of health. Based on the average calculation, the Euclid approach degree of Baoxing River is 0.39, suggesting that the entire river is in a moderate state of health.

### 5.2. Results Analysis and Discussion

Combined with the social and economic activities of towns in Baoxing County, the evaluation results are further analysed in order to find out the key factors affecting river health.

(1) According to the survey results of Baoxing River Basin, 14 indicators are of direct significance to judge the health of Baoxing river among the 15 indicators proposed by the study that can reflect the river health level. Based on the dynamic combination weighting technology composed of the DF method and entropy weight method, the biodiversity index of fish, water use intensity, endemic or indicative species retention, and COD emissions were identified as the key indicators, each weighted at over 10%. The biodiversity index of fish was weighted highest, at 27.4%. The total weight of the above four indicators is 67.9%, which determines the health level of Baoxing River. The biodiversity index of phytoplankton, biodiversity index of zooplankton and biodiversity index of benthic animal are three important indicators, which was weighted at 5%–9%. The total weight of them is 20.2%, which means that they are indicators that need more attention. The other seven indicators, with a total weight of 11.8%, have little impact on the health level of the river.

(2) The Euclid approach degree of Baoxing River was 0.39, indicating that the entire river is in a moderate state of health. The main reasons are as follows: ① The development of hydropower stations in upstream reaches results in the unstable supply of ecological flow in downstream reaches, and the biodiversity index of fish and endemic or indicative species retention are in low states. ② The mining in some towns and the aggregation of the urban population have led to high COD emission. Therefore, the main factors affecting the health level of Baoxing River are the development of hydropower and mining, followed by the discharge of domestic sewage. They should be the focus of future management. However, the result contrasts with the health state of rivers in China being mainly influenced by agricultural pollution and aquaculture pollution (according to the Chinese government report, the contribution of non-point source pollution to river health is close to 56%).

(3) Combined with the distribution of production and living activities of residents in Baoxing county (Figure 4), the health level and main influencing factors of different rivers reach are further analysed. ① Lingguan Town and Muping Town are urban areas of Baoxing County. They have the highest degree of urbanisation, with the most concentrated population, accounting for 51% of Baoxing County's population. At the same time, the construction scale and number of hydropower stations are relatively large in these regions, with COD emissions accounting for about 68% of the entire county. The Nemerow comprehensive pollution index is the most serious in these regions, resulting in the lowest level of biodiversity index of phytoplankton and biodiversity index of zooplankton, and poor river health. ② Daxi Town is located downstream of the Baoxing River Basin. Although the population is in seventh place in the county, the agricultural scale is in third place, which means that the river health of Daxi Town is mainly affected by agricultural production, livestock and poultry breeding, and upstream pollution. At the same time, there is no hydropower station in Daxi Town, so the water use intensity is low, and the development degree is low, resulting in the worst health of the river. ③ As the mining industry and hydropower stations are mainly distributed in Yongfu Town, Longdong Town and Wulong Town, the damage to the water quality and aquatic habitat in these towns is relatively serious, the water bodies there are in a moderate state of health. It is necessary to introduce green production technology to strictly control the sewage quality and improve the discharge flow of the hydropower station to ensure the ecological water demand of the river reach. ④ Qiaoqi Town and Fengtongzhai Town, with high altitude, are the water source regions of Baoxing River. The population density of them is relatively small, and the impact of domestic pollution sources is significantly lower than that of other towns. At the same time, due to the rich water system network, the large area of

natural water body, mining and hydropower development scale has not exceeded the water resources carrying capacity, and the river health in these areas is in a relatively good state. But in the future, mining and hydropower development urgently need to take more protection measures to contribute to the improvement of downstream river health.

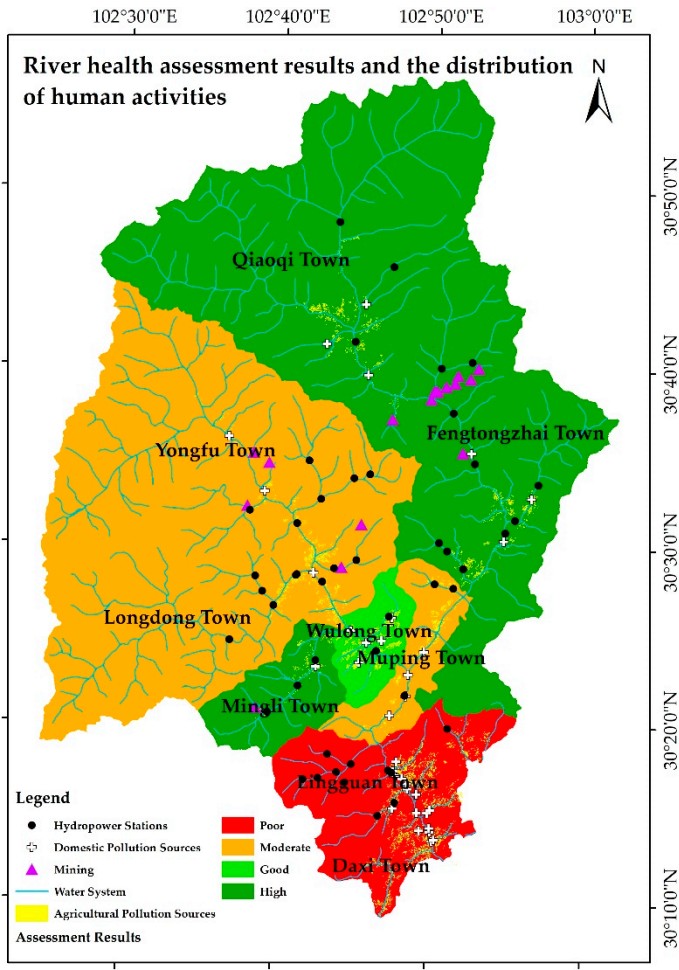

**Figure 4.** River health assessment results and the distribution of human activities.

The purpose of SDGs-based river health assessment for small- and medium-sized watersheds is to achieve the goal that protection is as important as development. Baoxing River is the primary source of the Qingyi River, a tributary of the Yangtze River. If the assessment is only for the purpose realizing the basic life of local people, no other production and development activities are carried out, and all developmental indicators related with hydropower development and mining are eliminated in the assessment process, the results obtained by using this study model are as follows:

$$
R_{\rho H} = \begin{bmatrix} & \text{Muping} & \text{Lingguan} & \text{Longdong} & \text{Fengtongzhai} & \text{Qiaoqi} & \text{Yongfu} & \text{Mingli} & \text{Wulong} & \text{Daxi} \\ & \text{Town} & \text{Town} & \text{Town} & \text{Town} & \text{Town} & \text{Town} & \text{Town} & \text{Town} & \text{Town} \\ \rho H_i & 0.48331 & 0.20084 & 0.36044 & 0.60897 & 0.65599 & 0.50939 & 0.57320 & 0.45555 & 0.26845 \end{bmatrix} \quad (21)
$$

The river health in the towns of Baoxing County can be listed from high to poor level: Qiaoqi Town, Fengtongzhai Town, Mingli Town, Yongfu Town, Muping Town, Wulong Town, Longdong Town, Daxi Town and Lingguan Town. The evaluation results show that the overall health level has been improved from moderate to good, especially in the middle section of Baoxing River, mainly because the adjusted indicators do not consider the potential contribution of a healthy river to local production and living activities. Therefore, how to effectively guide the development of water resources and the construction

of hydropower stations of appropriate scale according to the local water resources occurrence and hydrogeological conditions, and how to develop high-quality eco-tourism and eco circular agriculture which have less impact on the river ecosystem on the premise of ensuring the basic development demands of local people, are more in line with the requirements of sustainable development.

## 6. Conclusions and Policy Implications

(1) Based on the SDGs proposed for aquatic ecosystems, this study created a river health assessment index system for small- and medium-sized watersheds, consisting of 15 indicators in four areas: clean water, sanitation, the present status of biodiversity and threats to biodiversity. Human activities were comprehensively analysed based on a combination of protection and development.

(2) The dynamic combination weighting technology composed of the DF method and entropy weight method avoids the limitation of subjective influences of the traditional weighting method, and it can reflect the dynamic change process of different indicators' influence on the river health level, which can more truly reflect the contribution level of different indicators to the river health level. The fuzzy matter-element analysis model based on the multi-level and multi indicators breaks the limitation of traditional river health evaluation which focuses too much on water environment quality or water ecological quality, rather it establishes a comprehensive evaluation technology which integrates water ecology and water environment, and better judges the key issues and main measures that should be paid attention to in the future improvement of the health of the river ecosystem.

(3) The protection of life below the water should be enhanced. River health regulations are dependent on biodiversity. Zooplankton, phytoplankton and benthic animals have a relatively strong tolerance value and environmental adaptability. Therefore, in order to protect fish and indicative species, the habitat of underwater organisms can be protected by increasing the frequency of enhancement and releasing, and by saving energy and reducing emissions. At the same time, fish ladders or fish migration channels should be constructed. The discharge flow of hydropower stations should be comprehensively and scientifically determined. Refined 'one station, one policy' management should be implemented. Taking Baoxing County as an example, the proportion of water reduced river reach in Wulong Town and Muping Town exceeds 50%, and there is not enough ecological water to meet demand. With no space for power station construction, hydropower stations should not be built, and existing hydropower power stations should be more strictly controlled.

(4) On the basis of ensuring the living needs of indigenous people, the scale of human production and life should be effectively controlled, and the utilization efficiency of water resources and pollution control levels should be improved.

(5) SDGs-based river health assessment considers the importance of protection and development in the assessment process. It can be used to find the optimal combination of economic benefits and ecological benefits by comparing the cost of various interventions under different policy scenarios. Additionally, the government can make policies to realize the sustainable development of local natural-social ecosystems through this method.

**Author Contributions:** Conceptualization, C.X.; Formal analysis, C.X.; Funding acquisition, C.S.; Investigation, C.X., C.S. and S.C.; Methodology, C.X.; Project administration, C.S.; Resources, C.S.; Supervision, C.S.; Validation, S.C.; Writing—original draft, C.X. and S.C.; Writing—review & editing, C.S. All authors have read and agreed to the published version of the manuscript.

**Funding:** This work was supported by the National Key R&D Program of China under Contract No.2019YFC0507505.

**Acknowledgments:** The authors would like to thank the project team and Baoxing government for their full support and collaboration during the investigation.

**Conflicts of Interest:** The authors declare no conflicts of interest.

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
