# Peer review of "SDGs-Based River Health Assessment for Small- and Medium-Sized Watersheds"

_sustainability, doi:10.3390/su12051846_

Round 1

Reviewer 1 Report

There are some issues the authors should address:

Please provide an explanation of all abbreviations, when they appear in MS for the first time, e.g. COD in abstract

Few sentences are to general, e.g. line 37-39, or  needs rewriting, line 81-83

Please provide explanation and/or references for Nemero index, line 121

Line 265: there is no explanation why 2017 year was selected as a reference year

Results and discussion is the weakest part of the manuscript. It needs improvement

Reviewer 2 Report

Thank you for the opportunity to review the paper, "SDGs-Based River Health Assessment for Small- and Medium-Sized Watersheds," and for your patience while I completed the review. 

The purpose of this paper was to present a methodology for assessing river health with respect to the Sustainable Development Goals. It is novel both in its specific methods (fuzzy-element analysis) and in its theoretical basis (development-focused rather than striving to achieve ecological goals that omit human needs and influences). 

Overall I found the paper very interesting and well-written, but was left with a few very important questions. My opinion is that the study was done well, but the results were not surprising - downstream, populous subwatersheds were in poorer ecological health than upstream, rural areas. The policy recommendations stemmed from the particulars of the subwatersheds rather than being outcomes of the analysis itself. So, why is such a complex method necessary, when simpler methods point in the same direction, and the management needs are already known? I would like to see a stronger argument, particularly in the conclusions, why this methodology should be adopted in other places, in preference to simpler heuristics that are already available. Future work building on these results may be more interesting, for example by recalculating the indicators for different policy scenarios and comparing the cost of various interventions, but this might not be possible with the textual analysis.

Here are my line-by-line comments:

Notes from Xue et al, "SDGs-Based River Health Assessment for Small- and 2 Medium-Sized Watersheds"

Line 10, "county sustainable development" - should this be "country"? (Note: see comment on line 257, below.)

Line 52, please state Riparian, Channel and Environmental (RCE) so that the name of the method matches its acronym.

Line 62, nice use of "dialectical"

Line 99, this is an odd statement - "scientificity" and "conciseness" are not typical motivations for socioecological indices. Maybe try 'rigor' and 'parsimony'?

Table 1, would it be possible to format this so that words aren't split across lines?
I disagree with the assessment that a higher ratio of utilisation for water resources is a positive indicator. If you think of water use over a multi-year timeframe, demands are likely to be more stable than (or inversely proportional to) annual supplies, meaning that a higher average utilisation ratio can leave the system more vulnerable to drought.

Line 141, it's not clear what you mean by "deviation from the actual situation is easy due to the lack of subjective control." Please clarify.

Paragraph starting with Line 147, it's not at all clear what document corpus you would be analysing here; peer-reviewed papers? Reports? News articles? Blogs? Since this is unclear, it's hard to understand why you are turning to text analysis at all - isn't the actual content of the documents more important than their keyword counts?

Equation 5: What is 'lnm'? Is this the natural log of the number of objects to evaluate? If so, please use ln(m), or a space between them as in the ln pij portion of the equation.

Line 248, "county" again - I'm not sure this is a universal-enough level of administration to form the basis for river comparison; should this be "country"?

Line 257, Now I see where the 'county' comes in - it would have been easier if you had mentioned Baoxing County very early in the paper, in the abstract and introduction, as the context for the application of the methods.

Figure 3, please use actual numbers for the DEM, not just 'high' and 'low.'

Table 2, why is there a slash through the heading "Town Indicator"?

Line 308, please reformat this so that it doesn't take up a whole page.

Lines 310-312, this result is quite consistent with upstream reaches/upstream subwatersheds being in better ecological health than downstream reaches. The methods advanced in this paper are quite complex and require a lot of input data. Might it be possible to approximate these results with a simpler heuristic?

Lines 366-367, it seems as if this policy recommendation is a foregone conclusion, not a finding that truly relies on the metrics that are generated in this article. So why do you need all these rankings and indicators if it's already clear what should be done in each subbasin?

Reviewer 3 Report

The manuscript reports on a river health assessment index system taking into consideration the vulnerability characteristics of small- and medium-sized watershed ecosystems and focusing on both ecological protection and development behaviour. The adopted indicators describe the areas of clean water, sanitation, present status of biodiversity and threats to biodiversity. The river health assessment index is applied on the real case of the Baoxing River basin

The article is an original contribution and the topic is of interest for the readership of the Sustainability journal.

English language is clear and the presentation is good; anyway, I have detected some criticisms in the text that should be properly addressed.

The Authors can benefit from the comments below to improve their paper. These have to be accomplished before manuscript acceptance.

Abstract

The abstract is concise and reflects the content of the article.

Introduction

Aims of the study are properly clarified in the Introduction.

Lines 63-65: concerning the current river health assessment research only focuses on the natural health of water bodies and the human behaviour of pollution discharge … confined to the purpose of protecting the ecological environment, the Authors are recommended add some references. The following studies can be included as part of this introductory discussion:

  • Todeschini S., Papiri S., Ciaponi C. (2018). Placement strategies and cumulative effects of wet-weather control practices for intermunicipal sewerage systems. Water Resources Management, 32(8), 2885-2900, doi: 1007/s11269-018-1964-y.
  • Todeschini S., Papiri S., Sconfietti R. (2011). Impact assessment of urban wet-weather sewer discharges on the Vernavola river (Northern Italy). Civil Engineering and Environmental Systems 28(3): 209-229, doi:10.1080/10286608.2011.584341.

SDGs-based index system of river health assessment for small- and medium-sized watersheds

This section is clear and adequately detailed.

Figure 1and Table 1 are necessary and clear.

Lines 121-122: A reference could be added for the Nemero index.

Line 129: In Table 1 remove “the utilisation of” before “ratio of water resources”.

River Health Assessment Model

The presentation is clear and the proposed subdivision into paragraphs is effective. References are properly provided.

Result Analysis

I suggest to rename this section as “Results and Analysis”.

Line 255: I suggest moving the paragraph “Research area and data source” as a separate section before the “Results” section. Some more information on the Baoxing River Basin should be added (i.e., Total area of the basin, length of the water system, range of slope, …). Please also briefly describe the field sampling survey, data sorting           and statistical data analysis for the Baoxing County.

Legend of Fig. 1 should be checked. What does Value ‘High Low’ refer?

Conclusions and Policy implications

Conclusions seem reasonable and are supported by the results. Unfortunately, conclusions are relevant only for the Baoxing River Basin. I suggest the Authors to include a final sentence of more general value in order to promote the adopted approach for future studies on small- and medium-sized watershed ecosystems.

Line 355: I suggest to replace “27.406%” with “27.4%”.

Line 356-359: Perhaps, reduce the number of decimals for the values of the Euclid approach degree.

References

Two references are suggested in the “Introduction” Section on the impact of human pollution behavior on the habitats of aquatic organisms. One reference is suggested in the “SDGs-based index system of river health assessment for small- and medium-sized watersheds” Section for the Nemero Index. Apart from these references, based on my knowledge, no important reference is missing.

Round 2

Reviewer 3 Report

The manuscript has been significantly improved following the recommendations of the Reviewers; all my concerns have been addressed and convincingly justified.